# Caffeine for Prevention of Alzheimer’s Disease: Is the A_2A_ Adenosine Receptor Its Target?

**DOI:** 10.3390/biom13060967

**Published:** 2023-06-08

**Authors:** Stefania Merighi, Alessia Travagli, Manuela Nigro, Silvia Pasquini, Martina Cappello, Chiara Contri, Katia Varani, Fabrizio Vincenzi, Pier Andrea Borea, Stefania Gessi

**Affiliations:** 1Department of Translational Medicine and for Romagna, University of Ferrara, 44121 Ferrara, Italy; alessia.travagli@edu.unife.it (A.T.); manuela.nigro@edu.unife.it (M.N.); psqslv@unife.it (S.P.); cppmtn@unife.it (M.C.); chiara.contri@unife.it (C.C.); vrk@unife.it (K.V.); fabrizio.vincenzi@unife.it (F.V.); 2University of Ferrara, 44121 Ferrara, Italy; bpa@unife.it

**Keywords:** A_2A_ adenosine receptor, Alzheimer’s disease, amyloid beta, caffeine, neuroinflammation, therapeutic target

## Abstract

Alzheimer’s disease (AD) is the most prevalent kind of dementia with roughly 135 million cases expected in the world by 2050. Unfortunately, current medications for the treatment of AD can only relieve symptoms but they do not act as disease-modifying agents that can stop the course of AD. Caffeine is one of the most widely used drugs in the world today, and a number of clinical studies suggest that drinking coffee may be good for health, especially in the fight against neurodegenerative conditions such as AD. Experimental works conducted “in vivo” and “in vitro” provide intriguing evidence that caffeine exerts its neuroprotective effects by antagonistically binding to A_2A_ receptors (A_2A_Rs), a subset of GPCRs that are triggered by the endogenous nucleoside adenosine. This review provides a summary of the scientific data supporting the critical role that A_2A_Rs play in memory loss and cognitive decline, as well as the evidence supporting the protective benefits against neurodegeneration that may be attained by caffeine’s antagonistic action on these receptors. They are a novel and fascinating target for regulating and enhancing synaptic activity, achieving symptomatic and potentially disease-modifying effects, and protecting against neurodegeneration.

## 1. Introduction

Caffeine is one of the world’s most well-known and frequently consumed psychoactive substances. Its primary source is coffee, but it may also be found in tea leaves, guarana berries, and cacao beans. It’s important to note that other products that contain caffeine include gum, soft drinks, energy drinks, and prescription drugs [1,2]. It has been reported that the average daily consumption of caffeine worldwide is 76 mg, with greater intakes in the United States and Canada between 210 and 238 mg and more than 400 mg in Sweden and Finland [3]. The great popularity of coffee is a result of both its flavor and stimulating properties as well as the customs and cultures of many different nations. 

Caffeine exerts several biological effects through interaction with many molecular targets. At high concentrations, it inhibits molecules such as ryanodine receptors (RyRs), phosphodiesterase enzymes, and GABA_A_ receptors, and also protects against oxidative stress and exerts anti-inflammatory activity by decreasing pro-inflammatory and increasing anti-inflammatory marker levels [4,5,6,7,8]. More importantly, at low concentrations (250 µM), it acts as an antagonist of adenosine receptors (ARs) named A_1_R, A_2A_R, A_2B_R, and A_3_R, all expressed in neurons and glial cells, with the highest affinity for the A_2A_R. Caffeine prevents adenosine receptors from being activated by regulating brain functions such as sleep, cognition, learning, and memory. These effects may regulate brain diseases such as Alzheimer’s disease (AD), Parkinson’s disease (PD), Huntington’s disease (HD), epilepsy, pain/migraine, depression, and schizophrenia [9,10,11].

This review aimed to shed light on caffeine’s preventive properties in AD via A_2A_Rs, given that a significant amount of adenosine effects on cognition are exerted through this receptor subtype.

## 2. Caffeine Metabolism

Caffeine is a natural trimethylxanthine alkaloid with three methyl groups at positions 1, 3, and 7 (1,3,7-Trimethylxanthine) [12,13]. A single cup of coffee contains about 80–100 mg of caffeine leading to a peak blood concentration of 1 to 10 µM after 45 min of absorption through the small intestine [14]. In healthy adults, caffeine has a half-life of 3 to 7 h. It is primarily broken down into three dimethylxanthines in the liver by the cytochrome P450 oxidase enzyme system (CYP1A2 isozyme), including paraxanthine, theobromine, and theophylline, all of which have pharmacological effects on humans. A little portion (0.5–4.0%) of caffeine taken in is eliminated unaltered in the urine and bile, as well as in saliva, semen, and breast milk. Since caffeine is both lipid- and water-soluble, it may easily pass the blood-brain barrier and then act on the brain when inside.

Caffeine activates the central nervous system at low dosages, but at high blood concentrations, it can cause agitation, excitation, tremor, tinnitus, headaches, and sleeplessness. It has been reported that a smaller amount of caffeine (250 mg) had more favorable effects (exhilaration, tranquility, positiveness) than the larger dose (500 mg), which created more negative effects (tachycardia, agitation, stress) [15].

## 3. Alzheimer’s Disease (AD)

AD is a long-lasting neurodegenerative condition, projected to triple over the next thirty years reaching 135 million individuals, covering 50–70% of cases of neurodegenerative dementia [16]. This neurodegenerative illness, which is common in the elderly, is defined by a gradual and irreversible loss of cognitive function. The most typical AD indicators include memory loss, language difficulties, changes in mood, unwillingness, confusion, disorientation, and the loss of reasoning and judgment. Age, genetics, pre-existing conditions including diabetes mellitus, cardiovascular disease, diet, and lifestyle, which are hypothesized to play roles in its etiology, are some of the variables that may contribute to the advancement of AD [17]. There are two varieties of AD: “early onset forms” (ADAD, less than 5% of all cases) connected to mutations in the presenilin 1/2 (PS-1/2) and amyloid precursor protein (APP) genes and “late-onset AD” (LOAD), the most frequent sporadic form of the illness [18,19]. When opposed to sporadic LOAD, the symptoms of ADAD typically appear significantly earlier in life (46.2 versus 72.0 years). The pathology is characterized by the accumulation of pathophysiological events, such as amyloid peptide constituting senile plaques [20] and neurofibrillary tangles (NFT) composed of hyperphosphorylated tau [21]. In order to grade AD-related pathology, the ABC criteria, which are Amyloid, TAU (Braak stages), and neuritic plaque (CERAD) scores, are thus required [22,23,24]. For the “in vivo” definition of the pathology, a number of biomarkers have been created based on the neuropathological image. As a result, the ATN system (Amyloid-TAU Neurodegeneration) has been developed. It consists of three components: (1) estimate of the amyloid load (a reduction in the cerebrospinal fluid (CSF) and/or a cortical buildup at amyloid-PET of Aβ peptide); (2) pTAU cortical deposits at TAU-PET and/or a rise in pTAU in the CSF; and (3) a rise in total-TAU in the CSF, an atrophic pattern on brain MRI, hypometabolism on FDG-PET, or any combination of these findings evaluating the extension of neurodegeneration [25]. Both beta-amyloid (particularly the form Aβ_1-42_) and tau-hyperphosphorylated protein are brain-produced proteins that the brain is unable to get rid of. Many years before memory problems manifest, both of these proteins increase and begin harming neurons [26,27,28]. The hippocampus, a part of the brain, is where cell death starts. The temporal lobe’s hippocampus plays a major role in learning and memory functions. Cell death then spreads to affect the entire brain, resulting in further cognitive and functional issues that are seen in AD patients. It is still unclear what causes amyloid plaques and NFT to form [29]. It is known that they cause memory problems and behavioral disturbances by killing and damaging brain cells. β-amyloid oligomers, which may be neurotoxic such as amyloid plaques, are another possibility raised by the hypothesis. Additionally, the aberrant release of neurotransmitters such as glutamate has a role in the inflammatory and neuronal degeneration processes that occur inside the brain [30]. The complicated chain of events that leads to AD and its symptoms also includes the neuro-inflammatory process. The pathophysiology of AD does not stop with the neuronal system; it also includes microglial, astrocytic, and infiltrating cells from the peripheral nervous system, which contribute to neurodegeneration [31]. Therefore, the immune system has a major impact on the relationship between neurodegeneration and the neuroinflammatory process carried out by activated microglial cells [32]. The neuroinflammation may cause tau to misfold, become hyperphosphorylated, and produce amyloid-beta oligomers [33]. In addition, increased oxidative stress, neuronal inflammation, apoptotic cell death, and loss of synaptic integrity are additional mechanisms contributing to the pathogenesis of AD. 

Specifically, an important contribution to neurodegeneration is given by reactive oxygen species (ROS), created from a variety of sources, that produce mitochondrial malfunction, protein oxidation, lipid peroxidation, and DNA denaturation. The phospholipids in the brain’s membranes, in particular, are made of polyunsaturated fatty acids, which are more vulnerable to free radical damage [34] allowing hydrogen ions to be removed and thus causing lipid peroxidation, which is a significant characteristic in the neural system degeneration seen in AD [35]. Because brain oxidation can negatively impact enzymatic activities that are crucial for the physiological operations of neuronal and glial cells, protein oxidation is noticeably increased in AD cases. The DNA may be impacted by strand breakage and DNA-protein crosslinking as a result of the oxidation of some brain areas [36]. Therefore, ROS at high levels has destructive consequences and can play a key role in the degeneration of neural structures. Mitochondrial oxidative phosphorylation and ATP production are major sources of ROS [37]. A key regulator of the antioxidant response in cells is the nuclear factor (erythroid-derived 2)-like 2 (Nrf2) [38]. Nrf2 is retained in the cytoplasm when ROS levels are low. But when ROS generation rises, Nrf2 is translocated into the nucleus, where it promotes the transcription of a number of genes essential for the antioxidant response of the cells [39]. Furthermore, Nrf2 activation improves autophagy performance. In AD, however, the buildup of Aβ and tau lowers Nrf2 levels, lowering the antioxidant response. Because their autophagy-mediated turnover is hampered by the decreased Nrf2 levels, Aβ and tau continue to accumulate [40].

Nevertheless, despite the possibility that anti-inflammatory drugs might help prevent dementia, a Cochrane review of randomized controlled trials comparing aspirin or other NSAIDs with a placebo for the primary or secondary prevention of dementia failed to find any support for its use in treatment [41].

To date, the cholinesterase inhibitors donepezil, rivastigmine, and galantamine, as well as memantine, are being used as pharmacological treatments for AD patients to enhance synaptic acetylcholine (ACh) levels or decrease glutamate excitotoxicity [42,43,44]. The need for novel treatment approaches is critical because existing medications, however, only provide symptomatic relief and do not stop the spread of the disease.

To promote the removal of tau protein and amyloid peptide or to disrupt neurodegenerative biological mechanisms, many new pharmaceutical targets have been developed [45]. This is a difficult task, but a drug called Aducanumab that targets amyloid protein deposition has just hit the market, albeit its clinical usefulness is still debatable [46]. Yet, it is controversial if β-amyloid is unquestionably a successful target given the failure of multiple clinical studies with medications targeted at the Aβ protein [30]. 

In this setting, purinergic receptors, particularly those in the hippocampus, represent a novel and fascinating target for regulating and enhancing synaptic activity and achieving symptomatic, and perhaps disease-modifying effects.

## 4. A_2A_ Adenosine Receptors in AD

High amounts of endogenous purine nucleoside adenosine can control the production of excitotoxic mediators, restrict calcium influx, hyperpolarize neurons, and have modulatory effects on glial cells [47]. Increased neuronal activity, hypoxia, ischemia, or central nervous system injury can raise its concentration from 30–300 nM (physiological circumstances) to 10 µM or greater [48]. The actions of adenosine on the brain are primarily mediated via A_1_R and A_2A_R. A_2A_Rs are more common in the basal ganglia and in synapses, despite the fact that A_1_Rs are the most numerous and widely distributed. As a hub, A_2A_R activation reduces the effectiveness of pre-synaptic inhibitory systems, specifically A_1_Rs and cannabinoid CB_1_ receptors across the brain, and thus transforms pre-synaptic control from being inhibitory to facilitatory [49,50]. A_2A_Rs are also found in astrocytes and microglia cells, where they regulate Na^+^/K^+^-ATPase, glutamate uptake, as well as the generation of pro-inflammatory cytokines [51,52,53,54,55,56]. All of these outcomes may help to explain the selective A_2A_R-mediated modulation of synaptic plasticity [57].

A_2A_Rs in particular have received attention because of the significant influence they have on the cortex and hippocampus, particularly in relation to cognition, memory, and aging [58,59,60]. Long-term potentiation (LTP) in the hippocampus is facilitated by endogenous adenosine through the activation of A_2A_Rs. There is proof that A_2A_R directly affects NMDA receptors at the postsynaptic level via PKA and Src kinase-dependent pathways [61,62]. Additionally, A_2A_R increases AMPA-evoked currents, which has an effect on synaptic plasticity [63]. Specifically, A_2A_R’s capacity to modify synapses and pathways depends on and varies based on their co-localization. In particular, in the hippocampal CA3-CA1 synapse and in the striatum, A_2A_Rs co-localize with mGluR5 receptors and synergistically amplify NMDA effects [64,65,66,67,68,69]. It has also been proposed that A_2A_Rs function in concert with TrkB receptors, either through a cyclic AMP-mediated mechanism or through TrkB receptor transactivation, and therefore stimulate BDNF activities at synapses. Indeed, there may be a connection between BDNF effects and A_2A_R synaptic activity, and A_2A_Rs appear to be essential for BDNF-induced synaptic transmission, enhancement of LTP, as well as maintenance of normal BDNF levels in the CA1 region of hippocampal slices [70,71,72,73] (Figure 1 and Figure 2).

The hippocampus’ adenosine A_2A_Rs are now understood to be crucial for hippocampal-dependent plasticity and related cognitive processes. Accordingly, administration of the A_2A_R antagonist SCH58261 significantly reduces LTP in the hippocampus during associative learning [49,66,67]. In the presence of presynaptic A_2A_R overexpression a dysregulation of synaptic plasticity has been observed. It is interesting that pharmacological A_2A_R activation brought on by a brain injection of a specific agonist is enough to cause memory impairments [74]. Although A_2A_R expression is relatively low in the cortex and hippocampus of adult humans, it considerably increases during aging, which is marked by extensive synaptic remodeling. A_2A_Rs have been found upregulated in the brain of aged subjects as well as in the hippocampus of aged animals [75,76,77]. Such age-related changes in A_2A_Rs have also been found in the cortex/hippocampus of patients with AD. Original studies particularly pointed out that post-mortem samples from AD frontal cortex showed increased A_2A_R binding as compared to age-matched controls [78]. Recent results, in particular, demonstrate an increase in A_2A_R in the frontal white and grey matter regions, as well as in the hippocampus of AD patients compared to healthy participants, and particularly in AD subjects a notable overexpression in the hippocampus in comparison to the other locations has been reported. Interestingly, in peripheral platelets, an ideal model useful and accessible, used in neurobiological research to investigate the mechanisms of Aꞵ production in neurons, a similar trend was observed in AD patients in comparison to healthy subjects [25]. These findings have been confirmed in AD animal models [75,79,80,81,82,83,84,85]. Interestingly, antagonism of A_2A_R has been found to reduce amyloid-beta buildup in APPswe/PS1dE9 mice [83]. Another work in which the A_2A_R was removed from THY-Tau22 mice found that silencing A_2A_R protects the mice against tau-pathology-induced deficiencies in spatial memory and long-term hippocampal depression [86]. Overall, the findings suggest a contribution of A_2A_R to the pathogenesis of AD [87,88].

## 5. Caffeine and A_2A_R: In Vitro and In Vivo Models of AD

It has been pointed out that the primary strategy by which non-toxic amounts of caffeine operate on biological tissues and control activity in brain circuits is through the antagonism of ARs [89,90]. 

Caffeine’s structure is sufficiently similar to adenosine that it can connect to adenosine-specific receptors, which is crucial in understanding how caffeine operates in the human body. Literature data report that caffeine, following 4 months of administration, improves memory deficits and pathology in transgenic mice models of AD by reducing hippocampal Aβ contents through inhibition of beta- and gamma-secretase expression [79]. This protective effect was confirmed as well in “aged” APPsw mice that were already showing signs of mental decline, following 4–5 weeks of caffeine intake [91]. Additionally, acute or chronic caffeine treatment in various AD transgenic lines lowered both plasma and brain Aβ levels, but they were not related to cognitive function [92]. In the THY-Tau22 transgenic animal, chronic coffee consumption at an early pathogenic stage delayed the onset of spatial memory impairments and was linked with lower hippocampus tau phosphorylation [93]. Long-term caffeine treatment for 12 days prior to the administration of Aβ, when associated with acute caffeine before Aβ administration, in a mouse model of Aβ-induced cognitive decline, prevented cognitive loss through A_2A_R antagonism, whereas acute or subchronic treatment did not [94]. Furthermore, blocking the adenosine A_2A_R, which is synaptically localized in the hippocampus [95], prevents amnesia from being caused by Aβ [94], supporting the earlier finding that A_2A_R antagonists inhibit Aβ-induced damage in cultured neurons [96]. In rats with sporadic AD, induced by streptozotocin (STZ), acute caffeine treatment prevented memory decline, neurodegeneration, and A_2A_R increase [82]. 

The effects of caffeine and other AR antagonists on scopolamine-induced memory loss in Zebrafish were evaluated [97]. Interestingly, acute caffeine pre-treatment prevented scopolamine-induced amnesia in the inhibitory avoidance test [97]. Caffeine treatment also had no effect on social interaction or investigative assessment [97]. As evidenced by the neuroprotective effects of other blockers such as ZM 241385, DPCPX, dipyridamole, and EHNA against scopolamine-induced memory loss, adenosine blockade can generally prevent scopolamine-induced amnesia [97]. Interestingly, caffeine also prevented the short- and long-term memory impairment brought on by scopolamine in both people and mice [98,99].

It has been shown that caffeine or A_2A_R gene knockout in mice can ameliorate cognitive impairment by decreasing Tau-hyperphosphorylation provoked by traumatic brain injury (TBI) [100]. The findings suggest a novel post-TBI mechanism in which tau hyperphosphorylation results from A_2A_R activation, impairing memory, which may be restored by persistent coffee administration. Interestingly, long-term caffeine consumption by aged mice induced a downregulation of A_2A_R as well as reduced hippocampal damage [101]. 

Recent research has shown that caffeine consumption does not alter synaptic transmission in the hippocampus, prefrontal cortex, or amygdala. Instead, it increases the metabolism of synapses, which may be helpful in handling negative stimuli that cause brain dysfunction [102].

Numerous studies have emphasized the function of caffeine in controlling ROS, neuroinflammation, and other elements linked to the degeneration of neurons [103]. Caffeine treatment may improve memory problems in PS1/APP transgenic mice by modulating the production of brain-derived neurotrophic factor (BDNF) and tropomyosin receptor kinase B (TrkB) [104]. In a related study, it was shown that long-term coffee consumption may control the levels of BDNF glial fibrillary acidic protein (GFAP) in AD, hence controlling AD pathogenesis in the mouse brain [105]. It has been demonstrated, in a mouse model of synaptic dysfunction and neurodegeneration induced by lipopolysaccharide, that caffeine controls the levels of Nrf2 and TLR-4, which are involved in glial-cell-mediated neuroinflammation and apoptotic cell death in mice [106,107]. Similar outcomes were observed after administering caffeine to D-galactose-treated mice; these data demonstrated that caffeine significantly decreased inflammation induced by p-JNK, synaptic dysfunction, and memory impairment in mice. Similar results were obtained when D-galactose-treated mice were treated with caffeine; the findings showed that caffeine markedly reduced p-JNK-induced inflammation, synaptic, and memory dysfunction in mice [108]. 

In conclusion, it could be suggested that caffeine has the potential to modify the consequences of the etiology of AD by reducing neuroinflammation, oxidative stress, and the apoptosis of neurons (Table 1).

## 6. Caffeine and AD: Clinical Studies

The ability of caffeine to control cognition in AD has been the subject of several clinical research, some of the most pertinent of which has been summarized as follows.

Caffeine had a neuroprotective impact and was linked to a decreased risk of AD in a study of 54 AD patients who drank 73.9 ± 97.9 mg/day of caffeine over the 20 years before their diagnosis, compared to people without AD who consumed 198.7 ± 135.7 mg/day throughout the same 20 years of their lives [109].

A large-scale prospective study of older age onset of AD was carried out within the Canadian Study of Health and Aging [110]. People from 10 Canadian provinces were included in the study and 4615 of the 6434 eligible participants who were 65 or older in 1991 were still alive and took part in the follow-up research. When they answered a risk factor questionnaire in 1991, all subjects had a normal cognitive function. Five years later, their cognitive health was evaluated again using a comparable two-phase process that included a screening survey, followed by a clinical assessment where necessary. 3894 cognitively healthy controls and 194 AD patients participated in the analysis. Age, education level, and the apolipoprotein E 4 allele were all strongly linked to an elevated risk of AD while a lower risk of it was linked to the use of nonsteroidal anti-inflammatory medications, wine, coffee, and frequent exercise. Specifically, drinking coffee was associated with a 31% lower incidence of AD.

In addition, the relationship between caffeine consumption, cognitive deterioration, and incident dementia was investigated in a sample of people aged 65 and older from a population-based cohort collected from three French cities including 4197 women and 2820 men as participants. At baseline and at the two- and four-year follow-up, cognitive function, a clinical dementia diagnosis, and coffee intake were assessed. The results indicated that a minimum of three cups of coffee per day were linked to a reduced deterioration in verbal memory [111]. Women who consumed more than three cups of coffee per day showed less decline in verbal retrieval, and to a lesser extent in visuospatial memory over four years than women who consumed one cup or less. Age was shown to have an influence on the preventive effects of caffeine. Interestingly, there was no link between caffeine use and men’s cognitive impairment [111]. There have been studies showing that older men who drink coffee experience less cognitive deterioration than those who don’t. The least loss in cognitive function was shown to be related to three cups of coffee per day [112]. In a different study, 1409 adults over the age of 50 who drank three to five cups daily had a 62 to 70% reduced incidence of dementia and AD than non-coffee users (0–2 cups) [113]. In a different study, 23,091 Japanese adults over the age of 65 were used to assess the relationship between coffee drinking and incident risk of dementia. In these, 13,137 individuals were examined over a 5.7-year follow-up period, and 1107 incident dementia cases were found. Altogether, drinking coffee was strongly linked to a decreased incidence of dementia occurrences. Moreover, this statistically significant inverse connection was more striking in women, non-smokers, and abstainers of alcohol. Drinking coffee drastically lowers the chances of developing dementia [114]. According to data analysis from the National Health and Nutrition Examination Survey (NHANES) 2011–2014, which included a total of 2513 adults aged 60 or older, caffeinated coffee and coffee-derived caffeine were linked to cognitive function; however, decaffeinated coffee was not [115]. A placebo-controlled trial with 60 older individuals supported the aforementioned conclusions by demonstrating that there was no significant relationship between decaffeinated coffee and cognitive abilities [116]. Nevertheless, it was demonstrated in randomized, placebo-controlled research that decaffeinated coffee may safeguard cognitive function. Nonetheless, it should be highlighted that only a small number of male and female participants underwent the final analysis [117].

Recently, the Phase 3 clinical study NCT04570085 is examining the Impact of Caffeine on Cognition in Alzheimer’s Disease (CAFCA). The primary goal of the trial is to assess the effectiveness of caffeine on cognitive decline in AD dementia at the early to moderate stage as compared to a placebo (30 treatment weeks) (MMSE 16–24). The date Expected for the study’s completion is November 2024 [44]. 

The effects of caffeine and products containing caffeine must first be thoroughly studied since they may contain a wide range of bioactive chemicals that can change the main effects of caffeine [4,8]. Anyway, the role of caffeine in the enhancement of long-term memory consolidation in humans has been assessed by using a behavioral discrimination test following 24 h after administration [118]. Recently, it has been reported that regular caffeine consumption can have a long-term impact on neuronal activity and plasticity in the adult brain through coordinated actions on the epigenome, transcriptome, proteome, and metabolome. Specifically, coffee therapy increased the number of glutamatergic synaptic proteins in the hippocampus, which in turn improved the transcriptome controls in response to learning. These results promote the idea that frequent caffeine use allows the brain to more effectively store experience-related events [119]. The findings of research on caffeine’s impact on the nervous system and cognitive processes are highly encouraging, but further research is needed before including this drug in the care plan for persons with neurodegenerative disorders. 

## 7. AD, Caffeine, and Gender Differences

Age is a significant risk factor for AD, and women often live longer than males. One in five (20%) of women and one in ten (10%) of men are thought to have a lifetime risk of AD at age 45. However, the difference in longevity between men and women fails to fully clarify why women account for two-thirds of AD patients. Even when disparities in lifespan are taken into consideration, several studies have revealed that women are still at a greater risk [120]. Accordingly, women exhibited larger hippocampal shrinkage and a quicker deterioration in cognitive function when AD biomarkers such as Aβ1-42 cerebrospinal fluid levels and total tau were present [121]. According to a Mayo Clinic research on aging, the transition from moderate cognitive impairment (MCI) to AD was equivalent in men and women between the ages of 70 and 79, but greater in women after the age of 80 [122]. This is most likely due to differences in brain morphology between men and women, and it has been stated that males are predicted to endure more diseases since their heads are roughly 10% bigger and have higher brain capacity than women, a notion backed by autopsy data [123]. In summary, women had a greater incidence and prevalence of dementia than males, which might be attributed to a variety of variables, including women’s longer life expectancy, altered neuroanatomical functions but also the influence of estrogens as well as menopause and pregnancy [124,125].

Only a few research have looked at the influence of caffeine consumption on dementia incidence in the setting of gender differences. Caffeine use in white women aged 65–80 has been linked to a decreased risk of dementia or cognitive impairment [126]. Furthermore, consuming a moderate amount of coffee (3–5 cups per day) decreases the risk of dementia compared to not drinking coffee. In addition, the incidence of dementia was lower in women than in males, while no differences were observed for AD [113]. Caffeine has been shown in certain studies to have neuroprotective benefits, with women who consume more than three cups of coffee per day having less speech retrieval and spatial memory issues than women who drink less than one cup of coffee per day [111]. Moreover, caffeine’s psychostimulant component appears to decrease cognitive loss in women without dementia, particularly in the elderly. In summary, these studies reveal that caffeine consumption appears to lower the probability of acquiring dementia in both men and women, although further research is needed to determine if women benefit more than men. 

## 8. Conclusions and Perspectives

The fight against AD and its widespread social effects is gaining importance on a worldwide scale being this neurodegenerative disorder is still incurable and unavoidable despite the terrible impact it has on people’s quality of life and the enormous financial burden it places on healthcare systems, corresponding to 1% of the global economy. The universal effects of aging include a significant decline in neuronal activity and an increased vulnerability to neurodegeneration. This is seen in human populations as a growing frequency of AD. Every day, advancements in the fields of medicine and pharmacy may be seen. Modern research and diagnostic techniques have made it possible to diagnose numerous diseases quickly, apply tailored medication, and have a good probability of full recovery. Unfortunately, the rate of progress in neurological illnesses is slower than we would want. The multifactorial and complicated character of the majority of the diseases is the cause of the issue and AD is one of the neurological conditions that is more often researched. 

Caffeine may be neuroprotective against dementia and potentially AD, according to data reported in this review, which were based on “in vitro”, “in vivo” and clinical studies, although further research is required to fully understand the biological basis of this effect. It has been hypothesized that coffee intake may be a highly protective factor, possibly by acting through antagonism of A_2A_Rs damaging synaptic plasticity, memory, oxidative stress, and neuroinflammation. As a result, with the hope that a new A_2A_R antagonist can enter clinical practice for AD therapy, there is strong evidence that supports the use of caffeine as a preventive supplement to counteract and contain the memory decline as well as neuroinflammatory processes involved in neurodegenerative diseases.

## Figures and Tables

**Figure 1 biomolecules-13-00967-f001:**
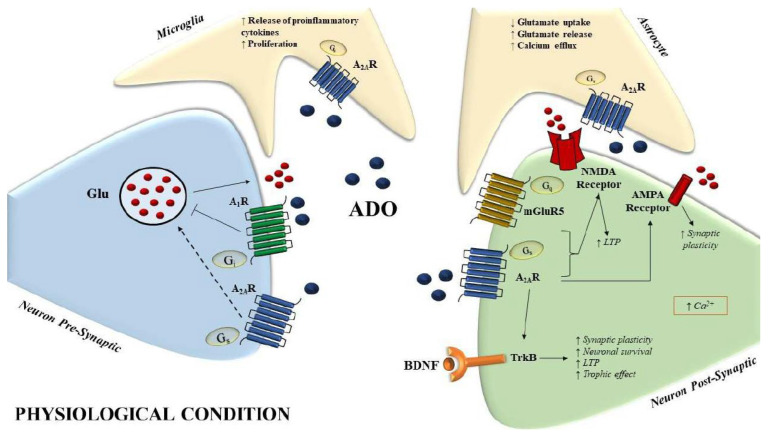
Schematic representation of A_2A_R effects in microglia, astrocytes, and synapses under physiological conditions. In astrocytes, it regulates the release and uptake of glutamate, and calcium efflux, while in microglia it increases pro-inflammatory cytokine secretion and proliferation. In the post-synaptic neurons, the A_2A_R increases synaptic plasticity and LTP through the modulation of glutamate NMDA and AMPA receptors and triggers the BDNF signaling pathway thus increasing neuronal survival. The short arrows with different directions indicate ↑(increase), ↓(decrease). Dashed line indicate a signal with minor strenght versus the solid line.

**Figure 2 biomolecules-13-00967-f002:**
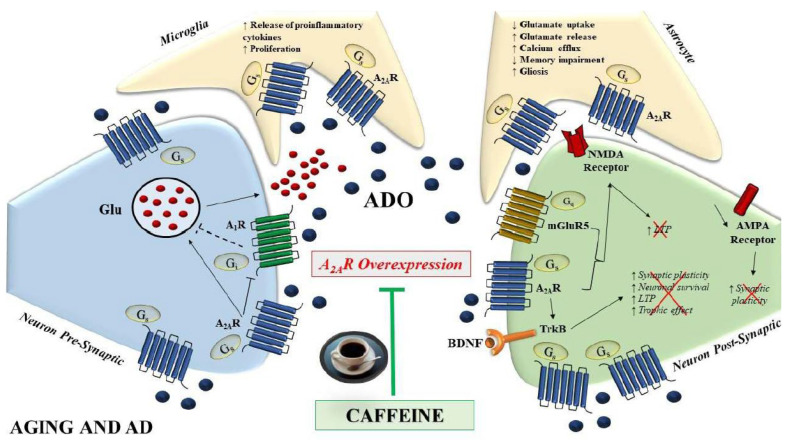
Schematic representation of A_2A_R effects in microglia, astrocytes, and synapses under AD pathology and aging conditions. A_2A_R expression is increased in neurons, astrocytes, and microglia, thus raising glutamate release, calcium efflux, and pro-inflammatory cytokine release, with a decrease in LTP and cognitive impairment. A_2A_R antagonism with caffeine attenuates memory disabilities and LTP alterations. The short arrows with different directions indicate ↑(increase), ↓(decrease). Red cross means signal not more active.

**Table 1 biomolecules-13-00967-t001:** Preclinical studies related to caffeine and A_2A_ adenosine receptors.

Experimental Model of AD	Main Results	Mechanisms of Neuroprotection	References
APPsw transgenic (Tg) mice SweAPP N2a neuronal cultures	Caffeine induces cognitive protection in spatial learning/reference memory, working memory, and recognition/identification tests	↓PS1 ↓BACE ↓Aβ ↓Aβ	[79]
Adult male Wistar rats with sporadic AD	Caffeine prevents Streptozotocin-induced memory decline, neuronal damage, and A_2A_R upregulation	ND	[82]
18–19 month-old APPsw mice	Caffeine improves working memory	↓PS1 ↓BACE1 ↓Aβ	[91]
AD Tg mice	Therapeutic value of caffeine against AD	↓Aβ	[92]
THY-Tau22 Tg mouse	Caffeine improves spatial memory	↓ Tau phosphorylation ↓ Proinflammatory and oxidative stress markers	[93]
CF1 adult mice injected with Aβ	Caffeine or selective A_2A_R treatment showed a protective effect against cognitive decline	ND	[94]
Primary rat cerebellar neurons	Caffeine shows neuroprotective effects through A_2A_R antagonism	↓Aβ-induced neuronal cell death	[96]
WT Zebrafish	Caffeine pre-treatment prevented scopolamine-induced amnesia	A_2A_R antagonism	[97]
Male CF1 mice	Caffeine increases memory and cognitive functions scopolamine-decreased	ND	[98]
A_2A_R KO mice	Caffeine normalizes memory impairment induced by TBI-mediated A_2A_R activation	↓Tau phosphorylation	[100]
Young and aged male Swiss mice	Caffeine decreases A_2A_R and the number of pyknotic aged neurons	ND	[101]
Adult male C57bl\6j mice	Caffeine increases the metabolic competence of synapses	ND	[102]
PS1/APP double Tg mice	Caffeine increases memory capability	↑BDNF ↑TrkB	[104]
Adult male albino Sprague-Dawley rats	Caffeine has a potentially good protective effect against AD induced by AlCl_3_	↑BDNF ↑TrkB	[105]
C57BL/6N male mice	Caffeine may regulate oxidative stress, neuroinflammation, and synaptic dysfunctions in LPS-injected mouse brains	↑Nrf2/TLR4	[106]
Adult male Sprague-Dawley rats	Caffeine has a beneficial effect against artificial senescence in mice induced by D-galactose	↓p-JNK, COX-2, NOS-2, TNFα and IL-1β	[108]

## Data Availability

Not applicable.

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
