# Peer review of "Caffeine for Prevention of Alzheimer’s Disease: Is the A2A Adenosine Receptor Its Target?"

_biomolecules, 2023, doi:10.3390/biom13060967_

Round 1
Reviewer 1 Report
The present review of the possibility that putative preventative effects of caffeine against Alzheimer’s disease (AD) are mediated by A2A receptors is clearly written and a good update of the evidence that those neuroprotective actions actually occur and could be used to treat or help treat AD.
There are, however, a few points that the authors may want to consider before publication:
-There are two “section 3”. The second one should be section 4 and, obviously, the subsequent sections should be renumbered.
-There are some statements or phrases that appear to be somewhat overstated or exaggerated, such as the phrase “Caffeine prevents adenosine receptors from being activated by regulating brain functions such as sleep, cognition, learning, and memory, and modifies brain diseases such as Alzheimer's disease (AD), Parkinson's disease (PD), Huntington's disease (HD), epilepsy, 43 pain/migraine, depression, and schizophrenia [9–11]” in the second paragraph of the Introduction.”[Introduction] There are some papers showing correlations and association of suspected caffeine intake with some reduced neurological or cognitive symptoms but none showing directly that caffeine modifies AD or other neurological disorders, so the statement could be moderated with this caveat. Also, those functions and disorders would be regulated by caffeine acting on the receptors and not the other way around.
-Likewise, the statement “Overall, the findings imply that the A2AR is essential for AD pathogenesis [83,84]” may not be accurate because despite evidence in animal models and cell cultures point to inhibition of adenosine receptors as a strategy for reducing cognitive deficits or reducing beta-amyloid deposition there is no good proof that those receptors are essential for AD pathogenesis in humans. Again, the statement may have to be more careful or provide direct evidence. The authors could say, for example, that the findings suggest a contribution of A2ARs to the pathogenesis of AD.
-There are other examples, (see last sentence of the “second” section 3) that could be reworded to ensure precise statements (caffeine may affect etiology, but it seems more likely that it modifies the consequences of the etiology of AD rather than changing the etiology itself).
- An important issue in discussing the cognitive effects of caffeine or coffee in human populations based in self-reports of coffee drinking is that any association of caffeine intake and cognitive impairment does not necessarily imply positive effects of coffee against cognitive impairment or AD, because it is also possible that as people develop progressive cognitive impairments tend to drink less coffee. That is, less coffee drinking might be a consequence of developing AD or MCI. Detailed attention to the evidence and specific studies may show that this is not the case. However, the authors may want to discuss the possibility of lower coffee drinking (or lower reporting of drinking) being caused by progressive loss of cognitive abilities or the ability to make decisions about coffee drinking.
Minor to moderate attention may be given to some expressions.
Author Response
The present review of the possibility that putative preventative effects of caffeine against Alzheimer’s disease (AD) are mediated by A2A receptors is clearly written and a good update of the evidence that those neuroprotective actions actually occur and could be used to treat or help treat AD.
There are, however, a few points that the authors may want to consider before publication:
-There are two “section 3”. The second one should be section 4 and, obviously, the subsequent sections should be renumbered.
We thank the referee and we have corrected accordingly.
-There are some statements or phrases that appear to be somewhat overstated or exaggerated, such as the phrase “Caffeine prevents adenosine receptors from being activated by regulating brain functions such as sleep, cognition, learning, and memory, and modifies brain diseases such as Alzheimer's disease (AD), Parkinson's disease (PD), Huntington's disease (HD), epilepsy, 43 pain/migraine, depression, and schizophrenia [9–11]” in the second paragraph of the Introduction.”[Introduction] There are some papers showing correlations and association of suspected caffeine intake with some reduced neurological or cognitive symptoms but none showing directly that caffeine modifies AD or other neurological disorders, so the statement could be moderated with this caveat. Also, those functions and disorders would be regulated by caffeine acting on the receptors and not the other way around.
We have modified the sentence accordingly.
-Likewise, the statement “Overall, the findings imply that the A2AR is essential for AD pathogenesis [83,84]” may not be accurate because despite evidence in animal models and cell cultures point to inhibition of adenosine receptors as a strategy for reducing cognitive deficits or reducing beta-amyloid deposition there is no good proof that those receptors are essential for AD pathogenesis in humans. Again, the statement may have to be more careful or provide direct evidence. The authors could say, for example, that the findings suggest a contribution of A2ARs to the pathogenesis of AD.
We have modified the sentence accordingly.
-There are other examples, (see last sentence of the “second” section 3) that could be reworded to ensure precise statements (caffeine may affect etiology, but it seems more likely that it modifies the consequences of the etiology of AD rather than changing the etiology itself).
We have modified the sentence accordingly.
- An important issue in discussing the cognitive effects of caffeine or coffee in human populations based in self-reports of coffee drinking is that any association of caffeine intake and cognitive impairment does not necessarily imply positive effects of coffee against cognitive impairment or AD, because it is also possible that as people develop progressive cognitive impairments tend to drink less coffee. That is, less coffee drinking might be a consequence of developing AD or MCI. Detailed attention to the evidence and specific studies may show that this is not the case. However, the authors may want to discuss the possibility of lower coffee drinking (or lower reporting of drinking) being caused by progressive loss of cognitive abilities or the ability to make decisions about coffee drinking.
We thank the reviewer for his/her interesting comment. However, we have not found in literature evidence about the possibility that it is AD progression that causes a minor intake of coffee. For this reason we have not added a comment in the review.
Reviewer 2 Report
Authors should include a table to summarize preclinical studies related to caffeine and adenosine receptors, including at least: animal and cellular model of AD, references, main results, mechanisms of neuroprotection...
Section 4. Caffeine metabolism should be moved to introduction
Authors should include a table to summarize clinical studies related to caffeine with the following columns: clinical trial number, phase, status, administration route, if completed indicate the date, main results, references
Include a paragraph about gender differences in Adinteladas the section 2 and try to justify differences in clinical studies according to sex.
Quality of figures 1 and 2 are low. Please modify them.
no comments
Author Response
Authors should include a table to summarize preclinical studies related to caffeine and adenosine receptors, including at least: animal and cellular model of AD, references, main results, mechanisms of neuroprotection…
We thank the reviewer for his/her suggestion, we have now added the table 1 reporting the studies requested.
Section 4. Caffeine metabolism should be moved to introduction
Ok, done
Authors should include a table to summarize clinical studies related to caffeine with the following columns: clinical trial number, phase, status, administration route, if completed indicate the date, main results, references
The reviewer asks to include clinical trials related to caffeine. The major part of clinical studies using caffeine are case-control studies, while we found only one RCT trial of phase III (NCT04570085) that is described in the text.
Include a paragraph about gender differences in Adinteladas the section 2 and try to justify differences in clinical studies according to sex.
We have added a new paragraph reporting both gender differences in AD and in AD following caffeine consumption.
Quality of figures 1 and 2 are low. Please modify them.
The figures 1 and 2 were created with the Powerpoint program and saved in JPEG format. Figures 1 and 2 are likely to degrade in quality after being placed into the text, anyway the original JPEG files will be obviously uploaded.